# A randomized controlled trial of pharmacist-led therapeutic carbohydrate and energy restriction in type 2 diabetes

Cody Durrer[1], Sean McKelvey[2], Joel Singer[3], Alan M. Batterham[4], James D. Johnson [2,5], Kelsey Gudmundson[1], Jay Wortman[6] & Jonathan P. Little [1,2✉]

Type 2 diabetes can be treated, and sometimes reversed, with dietary interventions; however, strategies to implement these interventions while addressing medication changes are lacking. We conducted a 12-week pragmatic, community-based parallel-group randomized controlled trial (ClinicalTrials.gov: NCT03181165) evaluating the effect of a low-carbohydrate (<50 g), energy-restricted diet (~850-1100 kcal/day; Pharm-TCR; n = 98) compared to treatment-as-usual (TAU; n = 90), delivered by community pharmacists, on glucose-lowering medication use, cardiometabolic health, and health-related quality of life. The Pharm-TCR intervention was effective in reducing the need for glucose-lowering medications through complete discontinuation of medications (35.7%; n = 35 vs. 0%; n = 0 in TAU; p < 0.0001) and reduced medication effect score compared to TAU. These reductions occurred concurrently with clinically meaningful improvements in hemoglobin A1C, anthropometrics, blood pressure, and triglycerides (all p < 0.0001). These data indicate community pharmacists are a viable and innovative option for implementing short-term nutritional interventions for people with type 2 diabetes, particularly when medication management is a safety concern.

[1] School of Health and Exercise Sciences, University of British Columbia, Kelowna, BC, Canada. [2] Institute for Personalized Therapeutic Nutrition, Vancouver, BC, Canada. [3] School of Population and Public Health, University of British Columbia, Vancouver, BC, Canada. [4] Centre for Rehabilitation, School of Health and Life Sciences, Teesside University, Middlesbrough, United Kingdom. [5] Diabetes Research Group, Life Sciences Institute, Faculty of Medicine, University of British Columbia, Vancouver, BC, Canada. [6] Faculty of Medicine, University of British Columbia, Vancouver, BC, Canada. ✉email: jonathan.little@ubc.ca

Type 2 diabetes is typically considered a chronic progressive disease, but it is now established that reversal/remission of type 2 diabetes is possible. Targeted nutritional approaches have garnered attention due to the increasing evidence base suggesting they can be used to induce type 2 diabetes reversal/remission[1–3]. In a non-randomized trial, continuous remote-care using a very low-carbohydrate, high-fat, ketogenic diet led to substantial weight loss, lowered haemoglobin A1C (HbA1c), and reduced need for glucose-lowering medications (including insulin) in a diverse group of 262 patients with type 2 diabetes[4–6]. In a cluster randomized controlled trial (RCT), Lean and colleagues[7] showed that a 12-week very low-calorie (~850 kcal day) total diet replacement method followed by food reintroduction resulted in remission of type 2 diabetes (sub-diabetes HbA1c and taking no glucose-lowering medications) in 46% of newly diagnosed patients at one-year follow-up. Online self-management interventions also report reduced oral diabetes medications and insulin dose, while lowering HbA1c, in participants with type 2 diabetes[8]. The idea that diet therapy could reduce or eliminate the need for glucose-lowering medications is intriguing but raises several important issues in diabetes care, including (1) how to limit the risk of hypoglycemia due to contemporaneous over-medication; and (2) the lack of guidance and/or knowledge of how to safely manage medication reductions when patients follow very low-carbohydrate or low-calorie diets.

While physicians are typically at the centre of diabetes care, pharmacists are more accessible and patients with type 2 diabetes make more annual visits to their pharmacist than primary care physician; this is especially true in rural areas[9]. Community pharmacists have expertise in medication management and can serve an important role in overall diabetes management[10]. Due to the need to reduce or eliminate glucose-lowering medications when type 2 diabetes patients follow a very low-carbohydrate or low-calorie diet[4,7,11], community pharmacists may be ideally positioned to safely and effectively deliver nutrition interventions targeted at reducing diabetes medication use and promoting type 2 diabetes remission. Accordingly, the aim of the Pharmacist-led therapeutic carbohydrate restriction (Pharm-TCR) as a treatment strategy for type 2 diabetes trial[12] was to determine if a very low-carbohydrate, low-calorie diet - led by community pharmacists - could reduce the need for glucose-lowering medications and facilitate improvements in cardiometabolic health when compared to guideline-based treatment-as-usual (TAU).

## Results

**Baseline characteristics of study participants**. Between July 7th, 2017 and April 1st, 2019, we recruited 188 individuals from 12 pharmacies across southern British Columbia, Canada. Sample size fell just short of the target recruitment of 100 per group; 98 participants randomized to the Pharm-TCR group and 90 participants randomized to the TAU group comprised the intention-to-treat population. Four participants in the Pharm-TCR group and 15 participants in the TAU group dropped out prior to commencing the trial. Furthermore, 16 participants in the Pharm-TCR group and 15 participants in the TAU group dropped out after commencing the trial. Sex assignment was missing for 13 participants. The CONSORT Flow Diagram is shown in Fig. 1. Baseline participant characteristics are reported in Table 1.

**Pharm-TCR led to greater discontinuation and reduction of glucose-lowering medication use**. At 12-weeks, 35.7% of participants in the Pharm-TCR group were completely off all glucose-lowering medications compared to 0% in the TAU group (absolute difference = 35.7%, 95% CI 25.9–44.8%, $p < 0.0001$). Within the Pharm-TCR group, 17.3% of participants achieved an HbA1c of <6.5% (i.e., below the diagnostic threshold for diabetes diagnosis) and were not taking any glucose-lowering medications compared to 0% in the TAU group (absolute difference = 17.3%, 95% CI 9.7–24.7%, $p < 0.0001$). Exploratory subgroup analyses for the primary endpoint of no medications by sex and insulin user status are presented in Supplementary Table 1. Changes in glucose-lowering medications and blood pressure-lowering medications, separated by medication class, are displayed in Supplementary Tables 3 and 4, respectively. To complement the binary primary outcome, the Pharm-TCR group had a lower mean medication effect score (MES) at 12 weeks ($p < 0.0001$; Table 2). Weekly MES in the Pharm-TCR group is displayed in Fig. 2.

**Pharm-TCR led to improvements in cardiometabolic health, anthropometrics, and health-related quality of life (HrQL)**. Secondary outcomes are reported in Table 2. Among clinical blood markers, HbA1c, fasting glucose, triglycerides, and GGT at 12 weeks were all lower in the Pharm-TCR versus TAU group (all $p < 0.0001$). Mean body weight, body mass index (BMI), waist circumference, body fat percentage, and systolic and diastolic blood pressure were also lower at 12 weeks in the Pharm-TCR group when compared to TAU (all $p < 0.0001$). For HrQL, measures of role functioning, mental health, health perceptions ($p < 0.0001$), and pain all improved in the Pharm-TCR group versus TAU. Descriptive outcomes assessed weekly in the Pharm-TCR group are displayed in Fig. 2. Mean daily macronutrient and kilocalorie intake at baseline, week 6, and week 12 are reported in Supplementary Table 5.

The exploratory statistical mediation analysis for the HbA1c (%) outcome revealed an indirect treatment effect (mediated via the change in body mass) of −0.8 (95% confidence interval: −1.3 to −0.3) percentage points ($P = 0.001$). This mediation effect represents 57% of the total causal effect of −1.4 percentage points (Table 2). The direct effect (not mediated by the change in body mass) was −0.6 (−1.2 to −0.04) percentage points ($P = 0.037$).

**Adverse events**. There were four adverse events reported in the Pharm-TCR group and no adverse events reported in the TAU group. Two of the adverse events were related to mild hypoglycemic events (recorded blood glucose levels of 4.1 mmol/L and 3.5 mmol/L); both events occurred when participants were reluctant to reduce insulin dosages by the recommended amount (one at the instruction of their endocrinologist) and were treated by the participants by consuming carbohydrates upon the advice of their pharmacist. Upon following the recommended medication adjustments, these participants reported no more hypoglycemic symptoms. One adverse event was related to reporting hypoglycemic symptoms; however, the participant recorded blood glucose values no lower than 5.2 mmol/L. The cause of these symptoms was suggested to be due to waiting too long between meals. Upon resolution of this issue, all hypoglycemic symptoms stopped. The final adverse event was a cardiac event that occurred three weeks into the study and was deemed not related to the intervention by the data and safety monitoring board.

## Discussion

There is mounting evidence that type 2 diabetes can be reversed through nutritional interventions. What must be considered now is how people with type 2 diabetes can access efficacious interventions and how healthcare practitioners can safely deploy them. This study provides RCT level evidence that community-based pharmacists can effectively and safely implement a dietary intervention that rapidly reduces the need for glucose-lowering

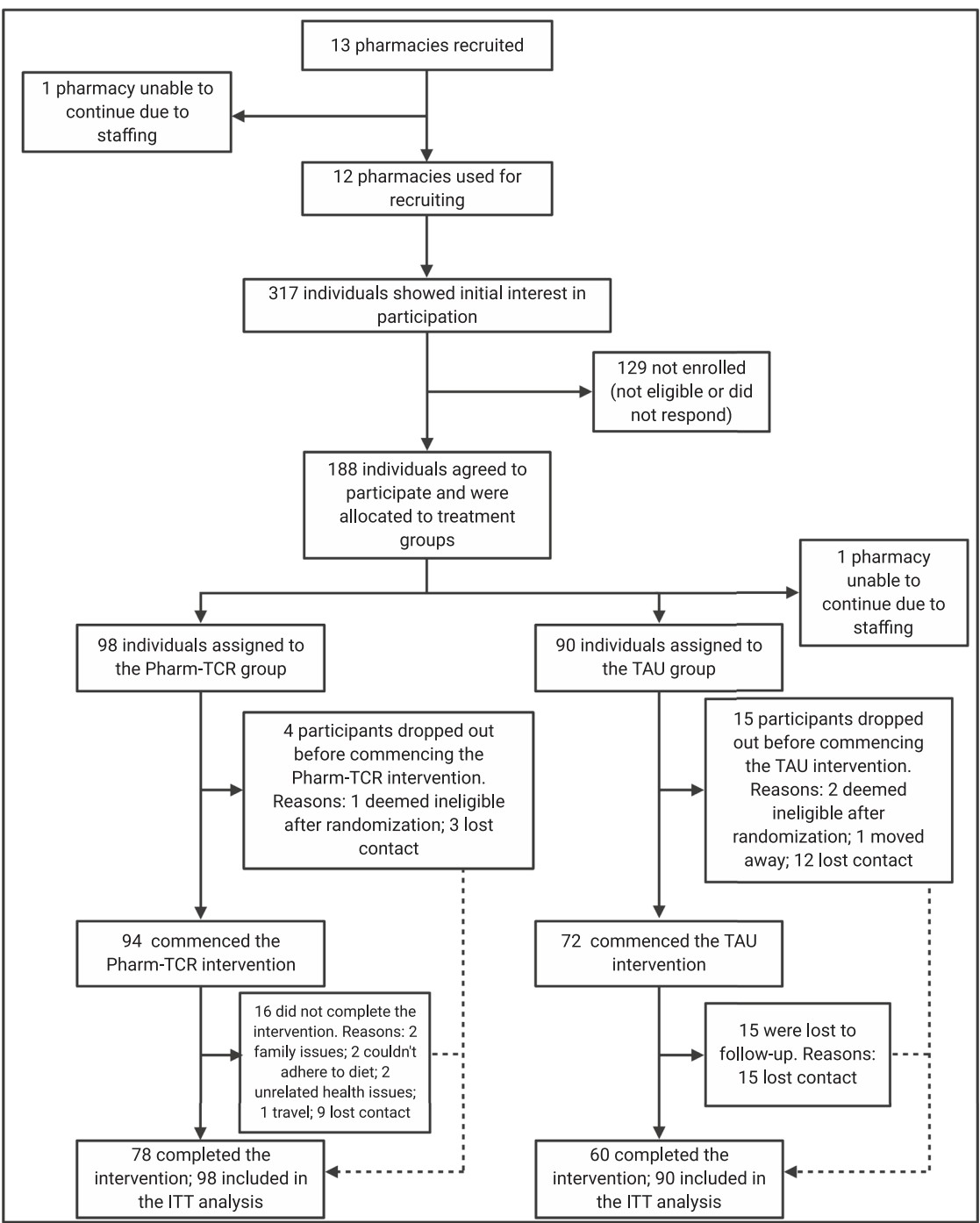

**Fig. 1 Trial CONSORT flow diagram.** Pharm-TCR Pharmacist-led therapeutic carbohydrate restriction, TAU Treatment-as-usual, ITT Intention-to-treat. Created with BioRender.com.

medications and improves cardiometabolic health in people with type 2 diabetes within a real-world setting.

When treating type 2 diabetes, standard clinical practice is to target glucose control and key risk factors that are associated with the development of micro- and macrovascular disease[13]. Cardiovascular disease accounts for ~33–50% of all diabetes-related deaths[14]. In this study, we demonstrate rapid improvements in glycemic control (while glucose-lowering medications were withdrawn or reduced), body weight, waist circumference, triglycerides, and blood pressure following the Pharm-TCR intervention. Furthermore, we report a significant reduction in GGT levels, which is correlated to reductions in excess liver fat[15]. Given

that excess liver fat is linked with liver insulin resistance[15], and therefore elevated fasting glucose levels, it is unsurprising that fasting glucose was substantially reduced in the Pharm-TCR group. Taken together, these data could indicate a depletion of harmful liver fat and a shift toward improved overall metabolic control. Interestingly, the sizeable reductions in blood pressure occurred despite reduced blood pressure medication usage in the Pharm-TCR group. The systolic blood pressure reductions in the Pharm-TCR group equate to a ~25% reduction in total cardiovascular mortality risk[16]. Post-intervention Pharm-TCR triglyceride levels and the reductions in body weight and waist circumference have also been linked to meaningful reductions in

**Table 1 Baseline characteristics.**

| | Pharm-TCR | TAU |
|---|---|---|
| **Sex** | | |
| Male (%) | 44 | 43 |
| Female (%) | 56 | 57 |
| Age (years) | 58 (11) | 59 (8) |
| T2D Duration (years) | 11.8 (8.0) | 8.8 (8.0) |
| Body weight (kg) | 102.3 (21.6) | 103.4 (19.7) |
| BMI (kg/m²) | 36.0 (6.0) | 35.1 (5.3) |
| Waist Circumference (cm) | 115.6 (13.6) | 115.8 (16.3) |
| Body Fat % (%; percentage points) | 39.0 (6.3) | 40.2 (6.3) |
| Systolic Blood Pressure (mmHg) | 138 (17) | 136 (16) |
| Diastolic Blood Pressure (mmHg) | 80 (14) | 82 (13) |
| HbA1c (%; percentage points) | 7.9 (1.5) | 7.8 (1.4) |
| HbA1c (mmol/mol) | 63 (16) | 61 (15) |
| Fasting Glucose (mmol/L) | 9.3 (3.2) | 9.3 (3.1) |
| GGT (U/L) | 40.8 (39.6) | 47.8 (42.6) |
| AST (U/L) | 24.1 (13.9) | 31.1 (21.5) |
| ALT (U/L) | 31.3 (20.0) | 42.8 (30.5) |
| HDL Cholesterol (mmol/L) | 1.21 (0.32) | 1.21 (0.31) |
| LDL Cholesterol (mmol/L) | 2.38 (0.95) | 2.49 (0.98) |
| Total Cholesterol (mmol/L) | 4.46 (1.15) | 4.56 (1.13) |
| Triglycerides (mmol/L) | 1.97 (1.47) | 2.0 (1.02) |
| hs-CRP (mg/L) | 5.1 (5.3) | 5.5 (5.1) |
| Medication Effect Score | 2.1 (1.7) | 1.8 (1.7) |
| Leisure Score Index | 22.2 (26.7) | 23.7 (21.7) |
| **Health Related Quality of Life** | | |
| Physical Functioning | 70.1 (28.8) | 69.6 (28.2) |
| Role Functioning | 71.9 (39.1) | 71.7 (40.7) |
| Social Functioning | 81.3 (27.9) | 84.4 (25.4) |
| Mental Health | 76.7 (16.6) | 74.4 (16.3) |
| Health Perceptions | 52.8 (24.4) | 49.2 (23.6) |
| Pain | 41.6 (27.0) | 47.6 (21.3) |

Data are mean (SD). *T2D* Type 2 diabetes, *BMI* Body mass index, *GGT* Gamma-glutamyl transferase, *AST* Aspartate aminotransferase, *ALT* Alanine aminotransferase, *HDL* High-density lipoprotein, *LDL* Low-density lipoprotein, *hs-CRP* high-sensitivity C-reactive protein

cardiovascular risk[17–19]. When combined with the significant reductions in glycemia, the Pharm-TCR intervention elicited widespread improvements that would be expected to markedly reduce the risk for both micro- and macrovascular diseases. Several HrQL variables also improved following the Pharm-TCR intervention, providing evidence of the enhanced quality of life. Collectively, these broad health improvements indicate that the Pharm-TCR intervention is treating the disease rather than just managing the resulting hyperglycemia. If these improvements are sustained over time, a reduction in risk for common co-morbidities of type 2 diabetes would be expected.

Medication changes in clinical trials of similar nutritional interventions (i.e., combined low-carbohydrate, energy-restricted diets) in type 2 diabetes are often not reported in great detail; however, Goday et al.[20] and Morris et al.[21] do report a reduction in medication use as descriptive or exploratory outcomes. The cardiometabolic health improvements observed in our trial are in line with reductions in HbA1c, weight loss, and changes in lipids in studies led by primary care nurses[21] and physicians[20]. Taken together, our findings suggest similar efficacy for treatment outcomes in low-carbohydrate, energy-restricted nutritional interventions implemented in community pharmacies.

The exploratory statistical mediation analysis for the HbA1c outcome showed that over half of the total mean treatment effect was mediated by the change in body mass. Gummesson et al.[22] reported a linear 'dose-dependent' relationship between weight loss and HbA1c reduction, with an estimated mean HbA1c reduction of 0.1 percentage points for each 1 kg of weight loss. In the current trial, the mean weight loss was approximately 12 kg, which suggests a mean reduction in HbA1c of around 1.2 percentage points - close to our observed point estimate of 1.4 percentage points. Nevertheless, only 57% of the mean reduction in HbA1c was mediated by weight loss in our trial. The direct effect of the intervention (not mediated by weight loss) could be due, in part, to carbohydrate restriction per se[23] although other aspects of glucoregulation (e.g., insulin sensitivity, beta-cell function) could be involved.

A specific strength of this study was the use of pharmacists to deliver the nutritional intervention in the community. The need for rapid medication adjustments (i.e., within days/weeks to avoid predictable medication-related events) when following a low-carbohydrate, low-calorie diet necessitates that someone knowledgeable in both type 2 diabetes and medication management has frequent and direct contact with participants. Given the risk of hypoglycemia and hypotension in this scenario, as well as the frequent visits that people with type 2 diabetes typically make to their local pharmacy[9], community pharmacists were uniquely positioned to fill this role. Although the intention was not for pharmacists to replace dietitians in delivering nutrition therapy, they were a suitable choice given the commercial weight loss plan that was selected to standardize the delivery of the nutritional intervention. This study highlights the potential of pharmacists in a multidisciplinary health team strategy that includes nutrition therapy. Given the current burden on primary care physicians in many countries, the lack of access to registered dietitians, and the fact that people with diabetes typically make 50% more visits to their pharmacists than their primary care physicians[9], future studies should investigate how a pharmacist-supported care model can be more broadly implemented.

Recently, studies have reported type 2 diabetes remission as an outcome[6,24]; however, the criteria used to define remission are often inconsistent between studies[25]. The American Diabetes Association[26] has previously suggested criteria for type 2 diabetes remission that was based primarily on the results of bariatric surgery studies. These criteria include the important caveat that remission of diabetes must demonstrate sustained improvements in glycemia in the absence of medications or ongoing therapy for at least one year. While a group of participants in this study did achieve normoglycemia while not taking glucose-lowering medications, the study's relatively short duration precludes defining this as type 2 diabetes remission. Recently, Taylor[27] used the term post-diabetes to describe individuals who had achieved sub-type 2 diabetes glycemic levels and ceased using glucose-lowering medications. This is fitting as it implies the person is no longer in a state of type 2 diabetes but may be at increased risk to redevelop type 2 diabetes due to potentially irreversible pathophysiological alterations that may have already occurred. The terms remission and reversal both imply that the pathophysiological alterations contributing to type 2 diabetes (i.e., beta-cell dysfunction and insulin resistance) have improved. Consequently, without directly measuring these parameters it would be impossible to use these terms. For this reason, we have chosen not to use these terms to describe our results. Whether the numerous beneficial improvements in cardiometabolic health observed with the Pharm-TCR intervention constitute type 2 diabetes reversal, type 2 diabetes remission, or classification of post-diabetes will require further study.

Although relatively short-term in duration, there are specific strengths of the study that should be highlighted. The trial was designed to be pragmatic in nature to allow for insights that could be beneficial for the implementation of similar interventions in the community. As such, having community pharmacists deliver

**Table 2 Secondary outcomes measured at 12-week follow-up.**

|  | Pharm-TCR | TAU | Treatment Effect | p value |
|---|---|---|---|---|
| Body weight (kg) | 91.9 | 103.9 | −12.0 (−13.6 to −10.4) | <0.0001 |
| BMI (kg/m$^2$) | 31.2 | 35.6 | −4.4 (−5.0 to −3.7) | <0.0001 |
| Waist Circumference (cm) | 102.4 | 113.8 | −11.4 (−13.1 to −9.7) | <0.0001 |
| Body Fat % (%; percentage points) | 35.0 | 38.6 | −3.7 (−5.0 to −2.5) | <0.0001 |
| Systolic Blood Pressure (mmHg) | 124 | 137 | −13 (−17 to −8) | <0.0001 |
| Diastolic Blood Pressure (mmHg) | 75 | 83 | −9 (−12 to −5) | <0.0001 |
| HbA1c (%; percentage points) | 6.4 | 7.8 | −1.4 (−1.8 to −1.0) | <0.0001 |
| HbA1c mmol/mol | 46 | 61 | −15 (−20 to −11) | <0.0001 |
| Fasting Glucose (mmol/L) | 7.2 | 9.1 | −2.0 (−2.9 to −1.1) | <0.0001 |
| GGT (U/L) | 19.5 | 26.9 | −27.6 (−38.5 to −14.7)%[#] | 0.00016 |
| AST (U/L) | 20.6 | 20.5 | 0.9 (−11.3 to 14.9)%[#] | 0.89 |
| ALT (U/L) | 24.7 | 26.0 | −5.0 (−19.7 to 12.6)%[#] | 0.55 |
| HDL (mmol/L) | 1.24 | 1.18 | 0.05 (−0.01 to 0.12) | 0.13 |
| LDL (mmol/L) | 2.42 | 2.24 | 0.17 (−0.04 to 0.38) | 0.106 |
| Total Cholesterol (mmol/L) | 4.14 | 4.22 | −0.07 (−0.36 to 0.19) | 0.60 |
| Triglycerides (mmol/L) | 1.00 | 1.49 | −34.3 (−43.4 to −23.7)%[#] | <0.0001 |
| hs-CRP (mg/L) | 3.0 | 3.5 | −12.4 (−30.3 to 10.2)%[#] | 0.26 |
| Medication Effect Score | 0.6 | 2.2 | −1.6 (−2.0 to −1.2)[†] | NA[†] |
| Leisure Score Index | 30.5 | 25.4 | 5.1 (−4.3 to 15.3)[†] | NA[†] |
| *Health Related Quality of Life* |  |  |  |  |
| Physical Functioning | 72.5 | 71.8 | 0.7 (−7.7 to 9.9)[†] | NA[†] |
| Role Functioning | 88.6 | 75.0 | 13.6 (2.4 to 26.3)[†] | NA[†] |
| Social Functioning | 93.8 | 87.8 | 6.1 (−2 to 14.3)[†] | NA[†] |
| Mental Health | 83.4 | 76.5 | 6.9 (1.9 to 12.7)[†] | NA[†] |
| Health Perceptions | 70.6 | 51.4 | 19.2 (13.2 to 25.4) | <0.0001 |
| Pain | 28.5 | 36.0 | −7.5 (−17.2 to −0.1)[†] | NA |

Data are adjusted means (Pharm-TCR & TAU) and effect estimates (Treatment Effect) and 95% confidence intervals derived from constrained baseline longitudinal analysis via linear mixed models. *P*-values are from two-sided tests. Secondary outcome *p*-values were not adjusted for multiple comparisons. [†] bias-corrected and accelerated confidence intervals derived from non-parametric bootstrap analysis. [#]Treatment effect expressed as a percent difference (ratio of geometric means) from log-transformed analyses (Pharm-TCR vs. TAU). 'NA', a precise *P* value cannot be obtained for a BCa bootstrap analysis. *T2D* type 2 diabetes, *BMI* body mass index, *GGT* gamma-glutamyl transferase, *AST* aspartate aminotransferase, *ALT* alanine aminotransferase, *HDL* high-density lipoprotein, *LDL* low-density lipoprotein, *hs-CRP* high-sensitivity C-reactive protein.

the intervention in an RCT design is a major strength of the study. Furthermore, using a standardized medication deprescription plan allowed for consistent implementation of the Pharm-TCR intervention in a safe and scalable manner.

Whilst not unexpected, participant attrition was a limitation to this study. Regardless, we treated dropouts in both groups as not achieving the primary outcome, so we feel the estimate of the effect of the intervention is robust. Furthermore, exploratory subgroup analyses suggested that the mean effect of the intervention was not substantially different between men and women, or insulin users versus non-users. However, we caution that confidence intervals for these sub-group effects are wide, as the trial was not powered for sub-group interactions. Finally, for continuous secondary outcomes we utilized a constrained baseline longitudinal analysis via a linear mixed model[28]. When there are no missing data, this model is equivalent to a standard regression model with baseline included as a covariate (ANCOVA). With missing data, the constrained baseline model is superior, as all participants with at least one measurement (baseline or post-intervention) are included in the analysis, given that baseline is part of the outcome vector. Thus, while we do not believe that the loss to follow-up negatively impacted the robustness of our results, we acknowledge that the attrition rate indicates that this type of intensive intervention delivered in the community, even while closely supported, might not be suitable for everyone.

Although this trial was designed to compare the Pharm-TCR intervention to usual care (i.e., TAU), some aspects of the design should be emphasized to ensure proper interpretation of the findings. The Pharm-TCR intervention was given to participants free of charge. It is possible that this could impact the translation of the results. Furthermore, although a comparison of cost-savings via medication use reduction vs. the cost to implement the intervention is an important question to be answered in future research, this study was not designed for cost-effectiveness analysis. Participants in the Pharm-TCR intervention also received more contact with the study personnel than participants in the TAU group. As the trial design was pragmatic in nature, the aim was to compare the Pharm-TCR intervention (which includes both the diet aspect and the increased monitoring) to usual care. As such, the outcomes in the Pharm-TCR group cannot be attributed entirely to the effects of the diet alone.

For a patient with type 2 diabetes, undertaking an impactful dietary change can be potentially dangerous if not properly informed and/or monitored by qualified healthcare personnel. The results of this study suggest that pharmacists can fill this role and can help to safely deprescribe glucose-lowering medications. Future research should investigate the durability of the cardiometabolic improvements observed and explore ways to optimize the delivery of therapeutic nutrition by incorporating community pharmacists into type 2 diabetes care teams.

The community pharmacist-led therapeutic carbohydrate- and energy-restricted dietary intervention effectively improved cardiometabolic health outcomes while safely reducing or eliminating glucose-lowering medications in patients with type 2 diabetes. Pharmacists could be viewed as an accessible and innovative option for implementing community-based and nutritional interventions for people with type 2 diabetes.

## Methods

**Study design and participants**. A pragmatic community-based RCT following a parallel-group design was conducted through 12 community pharmacies (independently owned within the same pharmacy banner) throughout southern British Columbia, Canada. Ethics approval was granted by UBC Clinical Research Ethics Board (H16-01539) and written informed consent was obtained from all study

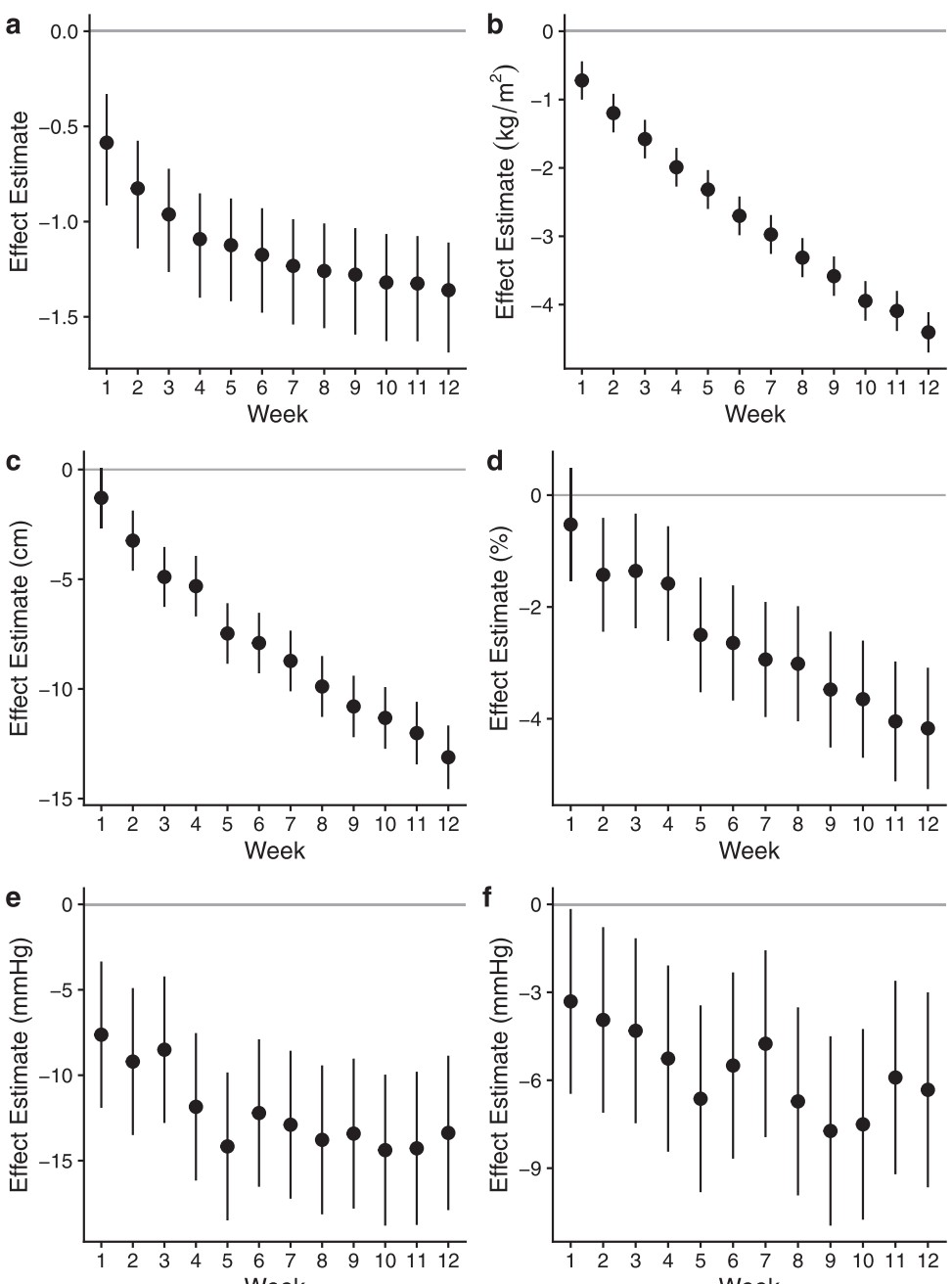

**Fig. 2 Descriptive analysis of weekly data collected in the Pharm-TCR group.** Data are weekly effect estimates for changes from baseline (Week 0; gray line) with confidence intervals in the Pharm-TCR group for (**a**) medication effect score (MES); (**b**) body mass index; (**c**) waist circumference; (**d**) body fat percentage; (**e**) systolic blood pressure; and (**f**) diastolic blood pressure. Values are effect estimates for adjusted mean change from baseline in **a**–**f**. Bias-adjusted and accelerated confidence intervals derived from non-parametric bootstrap analysis are presented in panel **a**. Error bars for panels **b**–**f** represent 95% confidence intervals. Data are based on participants for which baseline data were collected ($n = 92$) except for waist circumference ($n = 90$) and body fat percentage ($n = 91$). Source data are provided as a source data file.

participants prior to enrollment. Study conduct was performed in accordance with the ethical principles outlined in the World Medical Association Declaration of Helsinki. The trial was registered on ClinicalTrials.gov (NCT03181165) on June 8th, 2017 and the protocol detailing participant recruitment, study conduct, and planned data analyses is published elsewhere[12].

**Randomization and blinding.** Participants were randomized to either the pharmacist-led, therapeutic carbohydrate-restricted or treatment-as-usual control groups for 12 weeks. Randomization was performed by the pharmacist at each site through a secure password-protected website maintained by the Centre for Health Evaluation and Outcome Sciences (CHEOS). Random allocation was stratified by site (pharmacy) and glucose-lowering medications (≤2 vs. ≥3 or taking exogenous insulin) and performed on a 1:1 ratio using variable permuted block sizes.

Allocation lists were prepared using computer generation at CHEOS by a statistician unassociated with the study. Due to the nature of the trial, it was not possible to blind participants, study personnel at pharmacies, or research assistants to group allocation. The statistical analysis was also performed unblinded.

Participant inclusion criteria were: the ability to provide written informed consent, age 30–75 years, type 2 diabetes diagnosis by a physician, using at least one glucose-lowering medication, and a body mass index of ≥30 kg m[2]. Exclusion criteria included: history of a heart attack in the last two years; current unstable cardiovascular disorder; history of liver disease, kidney disease, or impaired renal function; currently pregnant, lactating, or planning to become pregnant within the next 12 months; diagnosed neurological disorder; history of bariatric surgery; history of cancer within the last five years; or dietary restrictions that would inhibit adherence to the intervention diet.

**Study procedures**. Potentially eligible participants were invited to the nearest participating pharmacy to review the eligibility criteria and study consent form. Recruitment was performed via a combination of newspaper and online advertising, posters posted in pharmacies, and word of mouth in each respective study site location. Prior to any data collection, written informed consent was obtained and participants' primary care physicians were notified of their participation. Upon admittance, eligible participants underwent baseline assessment of medication use, anthropometrics, blood pressure, and completed study questionnaires measuring HrQL and habitual physical activity (Godin Leisure-Time Exercise Questionnaire) at the pharmacy. A fasting blood sample was obtained by a local laboratory and participants completed a 3-day diet record to assess habitual food intake.

Participants in the Pharm-TCR group were asked to follow a commercial weight loss diet plan (Ideal Protein) supplemented with whole foods. The diet plan was provided free-of-charge and utilized a variety of low-carbohydrate, energy-restricted, adequate-protein meals and snacks that were used in combination with select meats and vegetables. Based on individualized food choices selected from within the diet plan, the daily macronutrient content equated to <50 g carbohydrates, ~35–45 g fat, and ~110–120 g protein for a total of ~850–1100 kcal. Participants in the Pharm-TCR group had weekly visits to the pharmacy to meet with a lifestyle coach and pharmacist to monitor progress, collect intervention foods, and assess medication usage. Lifestyle coaches managed scheduling and administrative tasks and were responsible for collecting anthropometric and blood pressure data while the pharmacists were responsible for all disease and medication data collection, information, and decisions. The medication deprescription plan is outlined in the supplementary information of the published protocol paper[12] (https://trialsjournal.biomedcentral.com/articles/10.1186/s13063-019-3873-7#Sec18). Data collected at weekly visits included: weight, height (visit 1 only), BMI, waist circumference, body fat % (bioelectrical impedance analysis), blood pressure, capillary blood ketones, and medication usage. On week 6 of the diet, participants completed another 3-day diet record. On the final visit, in addition to the typical weekly assessments, participants were assessed for the same blood, HrQL, diet record, and habitual physical activity measures that were collected at baseline.

Participants in the TAU group were given standard medication advice by their pharmacist as well as information pamphlets on diet and lifestyle conforming with 2013 Diabetes Canada (formerly the Canadian Diabetes Association) Clinical Practice Guidelines. Participants in the TAU group did not attend weekly meetings during the 12-week control period; however, they completed a 3-day diet record on week 6, similar to the Pharm-TCR group. Following the 12-week period, TAU participants returned to the pharmacy where they were assessed for the same blood, anthropometrics, HrQL, diet record, and habitual physical activity measures that were collected at baseline. Participants allocated to the TAU group were given the option of receiving the Pharm-TCR intervention after their initial 12-week TAU period in an effort to retain participants.

**Outcome measures**. The primary outcome measure was a binary outcome of either using or not using glucose-lowering medications after the 12-week study period. Secondary outcomes assessed at baseline and 12 weeks included HbA1c, change in glucose-lowering medication dose, BMI, body weight, waist circumference, body fat percentage, HrQL, blood lipid profile (total, HDL & LDL cholesterol, triglycerides), liver function tests (ALT, AST, and GGT), fasting plasma glucose, blood pressure, change in blood pressure medication dose, and a binary outcome of achieving both HbA1c <6.5% and no glucose-lowering medication use. Body weight and body fat percentage were assessed using the Tanita model DF-430 U (IL, USA), blood pressure was assessed using the PharmaSmart Model PS-2000C (BC, Canada), height was measured using the Seca model 700 (Germany), and waist circumference was assessed by measuring the distance around the waist at the top of the iliac crest with a tape measure. HbA1c, fasting plasma glucose, blood lipid profile, and liver function tests were analyzed by provincially accredited laboratories per standard clinical practice. The leisure score index was calculated by multiplying the number of strenuous, moderate, and mild bouts of physical activity by 9, 5, and 2, respectively[29]. HrQL was assessed using the 20-item Medical Outcomes Study Short Form (SF-20) and analyzed as described in Stewart et al.[30]. Medication effect score was calculated[31] as a continuous secondary outcome to quantify changes in medication use that were not captured by the primary outcome. Briefly, MES reflects the overall intensity of a diabetes medication regimen and is based on medication dosages and their efficacy for reducing blood glucose. Secondary outcomes assessed during the weekly visits in the Pharm-TCR group were MES, BMI, waist circumference, body fat percentage, systolic blood pressure, and diastolic blood pressure. Descriptive data for macronutrient content and energy intake from the 3-day food records were analyzed with MyFitnessPal (Under Armour®, Inc.).

**Statistical analysis**. All analyses were performed using R[32] and Stata software[33]. Data were analyzed as randomized. For the primary outcome, any participants lost to follow-up were presumed to be still using glucose-lowering medications. In Stata software, we applied a generalised linear model with a binomial distribution and logit link, with the Huber/White/sandwich variance estimator. To permit convergence, given zero events in the TAU arm, we specified the 'asis' command. The difference in proportion (Pharm-TCR vs. TAU) achieving 'no medications' and its asymmetric 95% confidence interval was derived using the -regpar- program[34]. The binary secondary outcome (both HbA1c <6.5% *and* 'no medications') was analysed with the same model. Exploratory subgroup analyses by sex and insulin user (yes/ no) were conducted by specifying a subgroup*treatment interaction. The exploratory subgroup

analyses for the primary outcome and the secondary outcomes of continuous HbA1c and body weight (by sex) are included in Supplementary Tables 1 and 2.

For the continuous secondary outcomes, missing data were assumed to be missing at random. Data were analyzed using constrained baseline longitudinal analysis via a linear mixed model[28], which allows all participants with at least one measurement (baseline or post-intervention) to be included in the analysis as a baseline is part of the outcome vector. The lme4 package[35] was used specifying Satterthwaite degrees of freedom, with fixed effects for timepoint (Pre vs. Post 12-weeks), treatment, sex, and the stratified allocation factors - study site and glucose-lowering medication use [2 or less, 3 or greater or taking exogenous insulin]) - and a random effect for participants to account for repeated measures within participants. 'Treatment' is a factor coded '1' if time = post and group = Pharm-TCR and coded zero otherwise. Subgroup analysis by sex was conducted for HbA1c and body weight by adding a sex*treatment interaction term. For descriptive purposes, outcomes assessed at the weekly visits in the Pharm-TCR group were analyzed using a linear mixed model with fixed effects for the week (0 to 12), sex, and the stratification factors (as above), and a random effect for participants.

The model specification was assessed visually using normal probability plots and residuals vs. fitted values plots. When the behaviour of the model residuals warranted a log transformation, effect estimates and 95% confidence intervals were back-transformed to ratio (percentage) differences using the emmeans package[36]. In cases where a log transformation could not be used (e.g. due to zero values), nonparametric bootstrap analyses were performed with 2000 resamples with replacement, and bias-corrected and accelerated 95% confidence intervals were calculated with the boot package[37,38].

We conducted an exploratory statistical mediation analysis using the Stata -sem- module, to examine the extent to which the observed treatment effect on HbA1c was mediated by the change in body mass. This analysis partitions the total causal effect into direct and indirect effects. The indirect effect is that passing through the putative mediator (change in body mass). The indirect effect/total effect × 100 gives the proportion (%) of the total treatment effect mediated by the change in body mass.

A sample size of approximately 100 per group was required to provide 80% power to detect a 20% difference (odds ratio of 2.67) in the proportion of patients on zero glucose-lowering medications (assuming 20% in TAU), with a two-sided P value of 0.05[39]. Throughout, 95% confidence intervals are interpreted as the plausible range of effect sizes compatible with the data and model.

**Role of the funding source**. The funder had no role in the study design, data collection, data analysis, data interpretation, or writing.

## Data availability

At the time of consent, it was not explicitly stated that data would be freely available in a public repository. As such, the corresponding author (J.P.L.) is the custodian of the data and will provide access to de-identified and processed participant data for academic purposes on request (jonathan.little@ubc.ca), with the completion of a data access agreement. Source data for Fig. 2 are provided as a Source Data file. Source data are provided with this paper.

## Code availability

Code used for statistical analyses is available at https://doi.org/10.17605/OSF.IO/76N28.

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

## Acknowledgements

The authors are grateful to the pharmacists and staff at each site for their hard work on carrying out this study. We are grateful to Dr Roger B Newson, Medical Statistician in the Department of Primary Care and Public Health, Imperial College London, UK, for his expert advice on the coding for the analysis of the primary endpoint including exploratory subgroup analyses. Peer-reviewed funding was obtained from the Mitacs Accelerate program (Grant No. IT08605). Matching funds for the Mitacs Accelerate fellowship to C.D. were provided by industry partner Pharmasave Drugs (Pacific) Ltd. Further funding support was provided through salary support to J.P.L. from the Canadian Institutes for Health Research (MSH-141980) and the Michael Smith Foundation for Health Research (Scholar Award #16890). Food products were provided in-kind to pharmacies by Ideal Protein. The funding agencies did not influence the study design or manuscript writing.

## Author contributions

The principal investigator J.P.L. and head pharmacist S.M. conceived the study and designed the intervention. J.P.L., S.M., and C.D. performed the literature search prior to the study design. J.P.L., S.M., J.S., A.M.B., and C.D. contributed to the study design. S.M., C.D., J.D.J. and J.W. were involved in data collection, C.D. oversaw day-today conduct of the study with support from J.P.L. and S.M. J.W. provided physician support. A.M.B., J.S., C.D., and K.G. performed the data analysis and statistical analysis. The underlying data were verified by C.D. and J.P.L. Figures and tables were created by C.D. J.P.L., J.D.J., A.M.B., and C.D. were involved in the interpretation of the data. The first draft of the manuscript was written by C.D. with guidance from J.P.L. All authors reviewed the manuscript, revised it critically for important intellectual content, and approved the final version.

## Competing interests

J.P.L. holds founder shares and advises for Metabolic Insights Inc., and is volunteer Chief Scientific Officer for the not-for-profit Institute for Personalized Therapeutic Nutrition. S.M. is employed as Chief Executive Officer for the not-for-profit Institute for Personalized Therapeutic Nutrition. J.W. is a member of the Scientific Advisory Board, and has received travel support and speaker's honoraria, from Atkins Nutritionals Inc. J.D.J. is Chair of the Board for the Institute for Personalized Therapeutic Nutrition and receives no compensation. C.D., J.S., A.M.B., and K.G. have nothing to declare.
