## [Peer Review File · Nature Communications]

Reviewer comments, first round –

Reviewer #1 (Remarks to the Author):

The clinical trial was clearly described and methods for analyses are reasonable. I do have some questions and comments concerning the representation of results.

1. Table 2: Although effects are displayed in % reduction from baseline to end-of-treatment, it would be good to also have absolute reductions from baseline to eot, as well as the observed correlation between these two. The authors may consider adding a table with supplementary values.
2. Table 2: I did not understand why the non-parametric bootstrap was not able to provide p-values but confidence intervals. Could these be calculated with a parametric bootstrap? How do results from non-parametric and parametric bootstraps compare?

Reviewer #2 (Remarks to the Author):

This study represents a short-term (12 week) randomised controlled trial examining the effects of a commercial low calorie, low-carbohydrate diet program delivered through pharmacy compared to usual care on diabetes medication requirements and other anthropometric and metabolic health risk biomarkers.

The primary outcome was that the commercial Pharm-TCR group achieved significantly greater medication reductions and improvements in metabolic health risk health markers.

This study makes a useful contribution to the field by demonstrating that pharmacy would potentially be a viable delivery pathway of a low-calorie, low-carbohydrate diet.

However, whilst the reported results are possibly a result of the structure and dietary composition of the Pharm-TCR program, a major limitation of the study was that the Pharm-TCR group received significantly greater professional support and product provisions, that was provided for free compared to the usual care group. This is a major limitation of the study, as it significantly limits that ability to draw major conclusions about the feasibility and efficacy of the Pharm-TCR program which users of this commercially available program are typically required to pay for these services and/or products. This limitation needs to be acknowledged and discussed in the manuscript.

Below are a number of the other comments and suggests and that would improve the manuscript.

Title:

Would be prudent to add the word 'short-term' to the title quality the duration of the study and intervention tested.

Abstract:

Include participant numbers in the abstract.

Describe the duration of the intervention.

Can numbers that quantify the changes in the medication usage (the primary outcome) be included?

How many participants in each group experienced medication discontinuation? Was this statistically significant? Please include information in abstract.

Include information that quantifies the reductions in HbA1c, anthropometrics, blood pressure and triglycerides for each group and the statistical comparisons.

Line 40: remove the word innovative, as this approach has been used previously so the level of innovation is substantially reduced.

Introduction

Line 48 – a new systematic review published by Goldenberg et al. in the BMJ discussing the role low carb diets for type 2 diabetes remission was published earlier this year and should be referenced.

Methods:

Line 263 – how were participants provided awareness of the study for the study invitation to be presented to attend the nearest pharmacy?

Line 267 – the blood pressure measurement should be listed here.

Line 319 - The binary secondary outcome (both HbA1c <6.5% and 'no medications') which seems to be referred to as diabetes remission was not identified as an outcome measure in the section above.

Line 325 – can it assumed that missed data were at random. Can this assumption be tested to confirm?

Can details of how the anthropometrics were specifically assessed be included? Were the instruments used to collect this information standardised across the pharmacies?

Can details of the dietary macronutrient composition of the TAU group be provided?

Given that risk of hypoglycaemia was discussed as an important issue in the introduction, was episodes or frequency of hypoglycaemic events assessed and can this information be reported? Only adverse events reported have been described, that does not necessarily mean additional hypoglycaemic events still occurred that were not sufficiently severe enough for participant to report it as an adverse event. This information is fundamental to understanding the feasibility of delivering nutrition interventions through pharmacy.

Results:

Please provide the reason for dropout in the consort flow diagram.

In consort diagram please indicate how many participants completed the study in each group.

Was there a statistical difference in dropout rate between the groups?

Can correlational analyses be performed to determine whether the changes in metabolic control are related to the degree of weight loss or possibly explained by other factors such as the reduction in carbohydrate content in the diet?

Line 96-100: The supplementary table provide raw data of changes by subgroup characteristics but do not provide any formal sub-group or sensitivity analyses. Can the results of these sensitive analyses be included.

Line 118-119 – Supplementary table 5 could not be viewed. The attached file only included a supplementary dataset. Can a summary table of the mean daily macronutrient and kilojoule intake of each treatment group at the key timepoint be included.

Discussion:

Line 139 – a major limitation of the efficacy of the study results is that participants were provided the Parm-TCR program free of charge. This should be acknowledged as a key component of the

community-based pharmacist approach. The study was not designed to test the efficacy of the approach when the user was required to pay for this commercial program and this should be stated as a key component of the program being delivered. Because of this design, how translatable are the results?

Line 39 – include the word 'short-term' as the long-term efficacy and safety still needs to be determined.

What are the estimated costs of delivering the Pharm-TCR program as implemented which would be useful to understand feasibility.

How were the pharmacy sites selected? Were they part of a single pharmacy chain which might limit the generalisability of the findings. This should be discussed.

The degree of professional support, supervision, food provisions and medical monitoring between the 2 groups was markedly different with the Pharm-TCR group being provided a significantly more food provision, support and medical monitoring which could at least in part explain the differences observed between the 2 groups. This needs to be discussed and acknowledged as a limitation of the study design. Would it have been possible that the TAU intervention would have experienced the same level of improvement if the same level of professional support (i.e. weekly visits), health weekly health monitoring, and food provisions were also provided?

Line 233 – the word 'should' be change to 'could' as all pharmacy practices are necessarily the same the relatively small pharmacy sample size limits wide scale generalizability of the findings.

Reviewer #3 (Remarks to the Author):

This manuscript presents the results of a pragmatic, community-based parallel-group randomized controlled trial aimed to evaluate the effect of a low-carbohydrate, energy-restricted diet (Pharm-TCR) compared to treatment-as-usual (TAU), delivered by community pharmacists on use of glucose-lowering medications, markers of cardiometabolic health, and health-related quality of life. Overall, it was demonstrated that the 12-week intervention effectively improved cardiometabolic health outcomes while safely reducing or eliminating glucose-lowering medications in patients with type 2 diabetes.

The manuscript is well-written, and the topic is highly interesting, given that community pharmacists could potentially have an active contribution in type 2 diabetes management. It is appropriate for publication provided that the authors would be willing to revise it according to the comments suggested below that will further contribute to its quality.

Methods:

- Lines 252-253: Was the person who conducted the statistical analyses aware of the group allocation?
- Lines 272-276: It is not clear whether the diet plan provided in Pharm-TCR was the same for all the participants or it was somehow personalized considering their individuals needs/ characteristics e.g. a range is mentioned for energy intake 850-1100 kcal/ day; does this vary considering individuals' age, sex, etc.? Perhaps this should be clarified.
- Lines 276-278: What was the role of the lifestyle coach? Was this a research assistant or is it part of the standard personnel in the community pharmacies? The contribution of the lifestyle coach is not discussed throughout the manuscript, given the importance of lifestyle change in diabetes management.
- The following points need to be briefly clarified in the method section: Who was in charge of collecting the data at baseline and 12 weeks, as well as at the weekly visits in the intervention groups? How was it secured that the same procedures followed throughout the different pharmacies/ study centers? Were the blood samples analyzed in accredited laboratories, following the same techniques? Which nutrient analysis tool was used to assess energy and macronutrient intake? The authors should also specify the type (brand, manufacturer etc.) of the equipment that were used measure blood pressure, body composition, weight, height, waist circumference.

Results:

- Lines 81-84: Which were the main reasons of dropout before commencing the trial and after commencing it?
- Table 1: Were there any statistically significant differences between groups at baseline for the characteristics presented in table 1? If yes, did you account for these differences while examining the intervention effect?

Discussion:

- The results of the present study should be briefly compared with the results of any relevant studies examining the effect of a low-carbohydrate, energy-restricted diet. Also, if data from other studies with longer follow-up are available, this could be also discussed, given the short duration of the current trial.
- Lines 138-141, 203-204: During the intervention period pharmacists were in weekly contact with study participants. Is this indeed feasible in clinical practice, within a real-world setting?
- Lines 166-167: Shouldn't this approach be combined with lifestyle coaching in order to achieve sustainable changes in diet and physical activity?

We thank the reviewers for their comments and constructive critiques. We have provided a point-by-point response to each reviewer comment in **blue type**. Relevant excerpts from the revised manuscript are in **red font**. References to the appropriate literature are provided as needed.

REVIEWER COMMENTS

Reviewer #1 (Remarks to the Author):

The clinical trial was clearly described and methods for analyses are reasonable. I do have some questions and comments concerning the representation of results.

1. Table 2: Although effects are displayed in % reduction from baseline to end-of-treatment, it would be good to also have absolute reductions from baseline to eot, as well as the observed correlation between these these two. The authors may consider adding a table with supplementary values.

Table 2 presents absolute values for the difference between groups for all variables except in the case of variables that required log-transformation. In the case of variables that required log-transformation to meet assumptions (GGT, ALT, AST, Triglycerides, and CRP), the values given in the second and third columns of Table 2 are the back-transformed geometric means in raw units. The difference between two geometric means is given by their ratio. For example, for GGT (mmol/L) the geometric mean is 19.5 in the intervention group and 26.9 in control. So, the difference is $19.5/26.9 = 0.72$; the GGT in the intervention arm at 12 weeks follow-up is 0.72x that in the control group, which equates to a treatment effect of -28%. This effect is the difference in outcome between the groups at 12-weeks, adjusted for baseline, sex, site, and the stratification factors. We feel that the data presented is the most appropriate and accurate statistical representation. The raw data and code is freely available per Nature Communications policy and guidelines. The reviewer can be assured that the effects displayed are not percentage reductions from baseline in each group.

2. Table 2: I did not understand why the non-parametric bootstrap was not able to provide p-values but confidence intervals. Could these be calculated with a parametric bootstrap? How do results from non-parametric and parametric bootstraps compare?

The BCa bootstrap confidence interval is generated directly from the empirical distribution whereas a precise p value may only be generated under a null hypothesis distribution. We prefer the non-parametric bootstrap, as the parametric approach relies on an arbitrary a priori choice of model - the nonparametric bootstrap is therefore considered more robust, especially for skewed distributions. The confidence interval gives the range of effects compatible with the data and model, and a P value provides no additional information in this case.

Reviewer #2 (Remarks to the Author):

This study represents a short-term (12 week) randomised controlled trial examining the effects of a commercial low calorie, low-carbohydrate diet program delivered through pharmacy compared to usual care on diabetes medication requirements and other anthropometric and metabolic health risk biomarkers.

The primary outcome was that the commercial Pharm-TCR group achieved significantly greater medication reductions and improvements in metabolic health risk health markers.

This study makes a useful contribution to the field by demonstrating that pharmacy would potentially be a viable delivery pathway of a low-calorie, low-carbohydrate diet.

However, whilst the reported results are possibly a result of the structure and dietary composition of the Pharm-TCR program, a major limitation of the study was that the Pharm-TCR group received significantly greater professional support and product provisions, that was provided for free compared to the usual care group. This is a major limitation of the study, as it significantly limits that ability to draw major conclusions about the feasibility and efficacy of the Pharm-TCR program which users of this commercially available program are typically required to pay for these services and/or products. This limitation needs to be acknowledged and discussed in the manuscript.

We thank the reviewer for their comments and constructive criticism. We have now acknowledged the greater attention and access to the pharmacist and commercial weight loss program in the trial with additions to the limitations section, as outlined below.

Below are a number of the other comments and suggests and that would improve the manuscript.

Title:

Would be prudent to add the word 'short-term' to the title quality the duration of the study and intervention tested.

Addressed, the title has been changed to explicitly state the length of the study:

Twelve weeks of pharmacist-led carbohydrate and energy restriction rapidly improves cardiometabolic health and reduces need for medications in type 2 diabetes: a randomized controlled trial

Abstract:

We have addressed most comments on the abstract but given the length constraints for Nature Communications abstracts (150 words) we were not able to provide all data and statistical comparisons. We have added the participant numbers, duration of intervention, primary outcome data and p-values for secondary outcomes.

Include participant numbers in the abstract.

Addressed

Describe the duration of the intervention.

Addressed

Can numbers that quantify the changes in the medication usage (the primary outcome) be included?

Primary outcome on medication usage discontinuation is now included.

How many participants in each group experienced medication discontinuation?

Now included per above.

Was this statistically significant? Please include information in abstract.

P-values are now included.

Include information that quantifies the reductions in HbA1c, anthropometrics, blood pressure and triglycerides for each group and the statistical comparisons.

Summary p-values for these secondary outcomes now included, abstract word limit (150 words) precludes inclusion of these data.

Line 40: remove the word innovative, as this approach has been used previously so the level of innovation is substantially reduced.

We respectfully disagree with this comment. We are unaware of any published trials or data on community pharmacists leading dietary interventions aimed at medication reductions in patients with type 2 diabetes. We agree that commercial weight loss programs have been tested/published, which is why we use the term “innovative” (implemented in a new way through community pharmacists) and not “novel” (the idea is not necessarily new).

Introduction

Line 48 – a new systematic review published by Goldenberg et al. in the BMJ discussing the role low carb diets for type 2 diabetes remission was published earlier this year and should be referenced.

This reference has been added. The reason it was not cited in the first submission is because we submitted this manuscript in October prior to the Goldenberg et al. review being published.

Methods:

Line 263 – how were participants provided awareness of the study for the study invitation to be presented to attend the nearest pharmacy?

Details of participant recruitment are in the protocol paper and have also been added to the “Study Procedures” section in the methods. The following sentence was added:

“Recruitment was performed via a combination of newspaper and online advertising, posters posted in pharmacies, and by word of mouth in each respective study site location.”

Line 267 – the blood pressure measurement should be listed here.

Added.

Line 319 - The binary secondary outcome (both HbA1c <6.5% and ‘no medications’) which seems to be referred to as diabetes remission was not identified as an outcome measure in the section above.

We have added explicit indication of this to the outcomes section. Given the uncertainties around definitions for type 2 diabetes remission we have been careful not to use this term to describe the results of our trial.

Line 325 – can it assumed that missed data were at random. Can this assumption be tested to confirm?

For the continuous secondary outcomes, we applied a principled approach to addressing missing data using a maximum likelihood approach. Like all such principled methods (maximum likelihood, full-information maximum likelihood, multiple imputation) it is assumed that the data are missing at random. There is no test that the data meet this assumption; one cannot know, for example, whether people with high HbA1c are more likely than those with low HbA1c to have missing data on HbA1c. However, we believe the assumption to be reasonable in this case. If these were primary outcomes, we could conduct sensitivity analyses to departures from the missing at random assumption by, for example, multiply imputing missing values and then adding or subtracting a clinically meaningful amount to indicate missing not at random – informative missingness – and then re-running analyses to examine the robustness of the results. However, for secondary outcomes this approach is neither necessary nor desirable, and the maximum likelihood-based approach is considered superior to a complete case analysis. In any event, there is evidence that the use of a method that is valid under a missing at random assumption can reduce bias even in the face of data that are missing not at random (Molenberghs et al., 2004).

Molenberghs G, Thijs H, Jansen I, Beunckens C, Kenward MG, Mallinckrodt C, et al. Analyzing incomplete longitudinal clinical trial data. *Biostatistics* 2004;5:445-64

Can details of how the anthropometrics were specifically assessed be included? Were the instruments used to collect this information standardised across the pharmacies?

This information, which is described in detail in the protocol paper, has now been added.

Can details of the dietary macronutrient composition of the TAU group be provided?

The macronutrient composition for the TAU group was not prescribed as they were given general information to follow Canadian Diabetes Association (now Diabetes Canada) guidelines, which does not explicitly prescribe macronutrients. Macronutrient composition for both the Pharm-TCR and TAU groups is provided in supplemental table 5. Per the comment below related to these data, there appear to have been issues accessing supplemental table 5 and we apologize if this was the case. We have re-uploaded this table (last page in the supplemental tables) and it appears in the manuscript pdf conversion/validation on our end.

Given that risk of hypoglycaemia was discussed as an important issue in the introduction, was episodes or frequency of hypoglycaemic events assessed and can this information be reported? Only adverse events reported have been described, that does not necessarily mean additional hypoglycaemic events still occurred that were not sufficiently severe enough for participant to report it as an adverse event. This information is fundamental to understanding the feasibility of delivering nutrition interventions through pharmacy.

Thank you for this clarification. Any hypoglycemic events reported by participants or study personnel are included in the subheading “adverse events” as reported. The hypoglycemic events reported by participants in the trial ended up not being “serious adverse events (SAEs)” but mild hypoglycemic symptoms as reported in the manuscript. Therefore, we believe what is reported (3 mild hypoglycemic events/symptoms) is the information the reviewer is requesting/referring to. We acknowledge the trial was not powered to detect hypoglycemic or adverse events, nor was it designed as a safety trial, but presenting this information provides evidence and context for the potential risks related to hypoglycemia with the intervention, as delivered.

Results:

Please provide the reason for dropout in the consort flow diagram.

Added.

In consort diagram please indicate how many participants completed the study in each group.

Added.

Was there a statistical difference in dropout rate between the groups?

It is not usual statistics practice to test the proportions lost to follow-up statistically. In the manuscript we state: “Four participants in the Pharm-TCR group and 15 participants in the TAU

group dropped out prior to commencing the trial. Furthermore, 16 participants in the Pharm-TCR group and 15 participants in the TAU group dropped out after commencing the trial.” Hence 20/98 dropped out in the treatment group vs. 30/90 in the control group. Differential drop out does not mean, necessarily, that results are biased and - on the flip side - the identical frequency of drop out in each arm does not indicate lack of bias (Bell et al., 2013). The two crucial factors are the presumed missingness mechanism and the analysis model. For the primary analysis we assume worst-case scenario – drop-outs are deemed not to have been successful in ceasing medication use. Hence, all randomized participants are included in the primary analysis in the groups to which they were originally assigned in accordance with the intention to treat principle. Our analysis is conservative and realistic. For the continuous secondary outcomes, as described earlier, we use a maximum likelihood approach to addressing missing data, based on a missing at random assumption. While there is no test for meeting this assumption, we believe it to be reasonable. In any event, there is evidence that the use of a method that is valid under a missing at random assumption can reduce bias even in the face of data that are missing not at random (Molenberghs et al., 2004).

Bell ML, Kenward MG, Fairclough DL, Horton NJ. 2013. Differential dropout and bias in randomised controlled trials: When it matters and when it may not. *BMJ (Clinical research ed)*. 346: e8668. doi: 10.1136/bmj.e8668.

Molenberghs G, Thijs H, Jansen I, Beunckens C, Kenward MG, Mallinckrodt C, et al. Analyzing incomplete longitudinal clinical trial data. *Biostatistics* 2004;5:445-64

Can correlational analyses be performed to determine whether the changes in metabolic control are related to the degree of weight loss or possibly explained by other factors such as the reduction in carbohydrate content in the diet?

This is an interesting question. The reviewer is referring to causal mediation analysis. The proper way to address this issue is via a statistical mediation model. This model partitions the total causal treatment effect – that reported in Table 2 – into the indirect effect (via the putative mediator or mechanism variable) and the direct effect (not mediated). We did not pre-specify mediation analyses in our protocol/ statistical analysis plan, but we have conducted such an analysis for the HbA1c outcome as an exploration. We feel that this analysis has enhanced the paper, so we thank the reviewer for raising the point. The results show that a little over half of the total causal treatment effect on HbA1c is mediated by the change in weight. We have uploaded Stata code for this analysis and added information to Methods, Results, and Discussion.

Line 96-100: The supplementary table provide raw data of changes by subgroup characteristics but do not provide any formal sub-group or sensitivity analyses. Can the results of these sensitive analyses be included.

Supplemental table 1 provides subgroup analysis of the primary outcome by sex and insulin use as specified in the methods. The proportion of participants that achieved the primary outcome in each group and the difference in proportions between the subgroups with the accompanying 95% confidence intervals are provided along with the relevant p-value.

Likewise, in supplemental table 2 the treatment effect on HbA1c and weight are provided for each sex along with the difference and relevant p-value. Therefore, we feel like we have provided formal subgroup and sensitivity analyses with 95% CIs and p-values as requested.

Line 118-119 – Supplementary table 5 could not be viewed. The attached file only included a supplementary dataset. Can a summary table of the mean daily macronutrient and kilojoule intake of each treatment group at the key timepoint be included.

We apologize for this issue. We have reuploaded supplemental table 5 and have provided it here for your convenience.

Supplemental Table 5: Macronutrient and kilocalorie intake at baseline, week 6, and week 12

	Kilocalories	Kilojoules	Carbohydrates (g)	Fats (g)	Protein (g)
Pharm-TCR					
Baseline	1789 (543)	7485 (2272)	188 (72)	72 (29)	85 (30)
Week 6	989 (217)	4138 (906)	69 (22)	34 (14)	106 (22)
Week 12	984 (212)	4116 (885)	66 (20)	34 (15)	106 (23)
TAU					
Baseline	1806 (608)	7558 (2544)	192 (80)	76 (32)	87 (28)
Week 6	1714 (597)	7172 (2497)	174 (71)	71 (37)	90 (28)
Week 12	1667 (589)	6975 (2465)	166 (85)	74 (45)	90 (28)

Data are daily means (SD). Descriptive data are based on complete cases of n=67 (Pharm-TCR) and n=53 (TAU).

Discussion:

Line 139 – a major limitation of the efficacy of the study results is that participants were provided the Parm-TCR program free of charge. This should be acknowledged as a key component of the community-based pharmacist approach. The study was not designed to test the efficacy of the approach when the user was required to pay for this commercial program and this should be stated as a key component of the program being delivered. Because of this design, how translatable are the results?

Thank you for this comment. Like most clinical trials (e.g., with a weight loss or glucose-lowering drug/therapy), we do not feel it would be ethically acceptable to ask participants to pay for the intervention/treatment being tested.

However, we agree with the reviewers point and have now acknowledged that for future translation and scale-up the cost-effectiveness of the intervention vs. savings in medications (along with the potential reductions in cardiometabolic risk and improved quality of life) would need to be assessed. This is included in the expanded limitations section.

Line 39 – include the word ‘short-term’ as the long-term efficacy and safety still needs to be determined.

This has been added.

What are the estimated costs of delivering the Pharm-TCR program as implemented which would be useful to understand feasibility.

Related to our response above we have included the cost of the intervention vs. drug savings as a limitation/future direction to the revised discussion. The study was not designed to test cost-effectiveness; therefore, attempting to conduct a such a post hoc analysis without purposeful data collection to answer this question could lead to misinterpretation. In a subsequent long-term trial we are planning to properly assess cost-effectiveness and incremental cost utility ratios appropriately with preplanned data collection and inclusion of an expert in health economics/cost-effectiveness analyses.

How were the pharmacy sites selected? Were they part of a single pharmacy chain which might limit the generalisability of the findings. This should be discussed.

Yes, per the protocol paper all sites were independently owned but from the same national pharmacy banner. This has been added to the methods.

The degree of professional support, supervision, food provisions and medical monitoring between the 2 groups was markedly different with the Pharm-TCR group being provided a significantly more food provision, support and medical monitoring which could at least in part explain the differences observed between the 2 groups. This needs to be discussed and acknowledged as a limitation of the study design. Would it have been possible that the TAU intervention would have experienced the same level of improvement if the same level of professional support (i.e. weekly visits), health weekly health monitoring, and food provisions were also provided?

We agree with the reviewer's comment and have expanded upon this in the limitations and discussion. As the trial was designed to compare the Pharm-TCR intervention to usual care, the purpose of the study was not to compare to an equal contact comparator using a different dietary intervention.

Line 233 – the word 'should' be change to 'could' as all pharmacy practices are necessarily the same the relatively small pharmacy sample size limits wide scale generalizability of the findings.

Addressed.

Reviewer #3 (Remarks to the Author):

This manuscript presents the results of a pragmatic, community-based parallel-group randomized controlled trial aimed to evaluate the effect of a low-carbohydrate, energy-restricted diet (Pharm-TCR) compared to treatment-as-usual (TAU), delivered by community pharmacists on use of glucose-lowering medications, markers of cardiometabolic health, and health-related quality of life. Overall, it was demonstrated that the 12-week intervention effectively improved

cardiometabolic health outcomes while safely reducing or eliminating glucose-lowering medications in patients with type 2 diabetes.

The manuscript is well-written, and the topic is highly interesting, given that community pharmacists could potentially have an active contribution in type 2 diabetes management. It is appropriate for publication provided that the authors would be willing to revise it according to the comments suggested below that will further contribute to its quality.

Methods:

- Lines 252-253: Was the person who conducted the statistical analyses aware of the group allocation?

The statistical analysis was not performed blind; this has been added. We do not feel that there is a substantial risk of bias here, given that the analyses were conducted in line with the protocol and statistical analysis plan, with any deviations clearly explained. Moreover, raw data and analysis code are freely available to facilitate reproducibility efforts.

- Lines 272-276: It is not clear whether the diet plan provided in Pharm-TCR was the same for all the participants or it was somehow personalized considering their individual needs/ characteristics e.g. a range is mentioned for energy intake 850-1100 kcal/ day; does this vary considering individuals' age, sex, etc.? Perhaps this should be clarified.

The range for energy intake (~850-1100 kcal/day) is based on the individual choices made for food options in the diet plan available to the participants. This reflects the participants' choice of different Ideal Protein foods (e.g., selecting a protein shake vs. cereal for breakfast) and the type of protein they choose to supplement their dinner meal with (e.g., salmon vs. chicken), based on their personal preference. There were no specific considerations made for age, sex, etc. We have added clarification to the methods.

“Based on individualized food choices selected from within the diet plan, the daily macronutrient content equated to <50g carbohydrates, ~35-45g fat, and ~110-120g protein for a total of ~850-1100 kcal.”

- Lines 276-278: What was the role of the lifestyle coach? Was this a research assistant or is it part of the standard personnel in the community pharmacies? The contribution of the lifestyle coach is not discussed throughout the manuscript, given the importance of lifestyle change in diabetes management.

We thank the reviewer for this comment, which we agree requires clarification. Lifestyle coaches were not study personnel and were not a special addition for the conduct of the trial. All pharmacies in the study employ a lifestyle coach to help manage the logistics of scheduling and administering the commercial weight loss diet plan, primarily for pragmatic reasons. We have added clarification to the methods and included this in the discussion for accuracy. It is likely that lifestyle coaches could be an important adjunct to diabetes and pharmacist-based management in this context.

The following has been added to the methods:

“Lifestyle coaches managed scheduling and administrative tasks and were responsible for collecting anthropometric and blood pressure data while the pharmacists were responsible for all disease and medication data collection, information, and decisions.”

- The following points need to be briefly clarified in the method section: Who was in charge of collecting the data at baseline and 12 weeks, as well as at the weekly visits in the intervention groups?

Please see our response and addition to the methods in the previous comment. The same lifestyle coach at each site was responsible for collecting the anthropometric and blood pressure data. The pharmacist was responsible for reviewing the medication use and developing/communicating the medication plan to the participant and physician as well as overseeing the safe implementation of the plan.

How was it secured that the same procedures followed throughout the different pharmacies/ study centers?

The lifestyle coaches undergo standardized training to deliver the commercial weight loss program and had additional study specific training via online training videos to ensure consistency in study conduct across sites.

As outlined in the published protocol paper, all pharmacies/sites underwent online training videos on study procedures and had a site visit by the research coordinator prior to the study starting.

Were the blood samples analyzed in accredited laboratories, following the same techniques?

Yes, these were made by provincially-accredited laboratories per standard clinical practice. This information has been added.

Which nutrient analysis tool was used to assess energy and macronutrient intake?

This information has been added.

The authors should also specify the type (brand, manufacturer etc.) of the equipment that were used measure blood pressure, body composition, weight, height, waist circumference.

This information, originally in the published protocol paper, has now been added.

Results:

- Lines 81-84: Which were the main reasons of dropout before commencing the trial and after commencing it?

The CONSORT flow diagram has been updated to reflect all reasons obtained for dropouts.

- Table 1: Were there any statistically significant differences between groups at baseline for the characteristics presented in table 1? If yes, did you account for these differences while examining the intervention effect?

It is not regarded as best practice to conduct null hypothesis significance tests for baseline values in randomized controlled trials as any difference between groups is, by definition, due to chance. There have been many statistical commentaries published arguing to eradicate statistical testing for baseline differences from the literature. Thus, we feel it is inappropriate to conduct a null hypothesis statistical test on baseline differences. See, for example:

de Boer, M.R., Waterlander, W.E., Kuijper, L.D. *et al.* Testing for baseline differences in randomized controlled trials: an unhealthy research behavior that is hard to eradicate. *Int J Behav Nutr Phys Act* 12, 4 (2015). <https://doi.org/10.1186/s12966-015-0162-z>

Senn SJ. Testing for baseline balance in clinical trials. *Statistics in Medicine* 1994; 13:1715–1726.

Furthermore, the CONSORT 2010 statement for the reporting of parallel group randomised controlled trials states:

“Tests of baseline differences are not necessarily wrong, just illogical. Such hypothesis testing is superfluous and can mislead investigators and their readers. Rather, comparisons at baseline should be based on consideration of the prognostic strength of the variables measured and the size of any chance imbalances that have occurred.”

And:

“In RCTs, the decision to adjust should not be determined by whether baseline differences are statistically significant.”

For the analysis of the continuous secondary outcomes, the baseline value of the outcome is constrained to be equal between arms, and sex and the stratification variables are included in the model. Our analyses are robust with results presented in line with CONSORT.

Discussion:

- The results of the present study should be briefly compared with the results of any relevant studies examining the effect of a low-carbohydrate, energy-restricted diet. Also, if data from other studies with longer follow-up are available, this could be also discussed, given the short duration of the current trial.

We agree and have added comparison to other studies for additional context. The following paragraph has been added to the discussion.

Medication changes in clinical trials of similar nutritional interventions (i.e., combined low-carbohydrate, energy-restricted diets) in T2D are often not reported in great detail; however, Goday et al. and Morris et al. do report a reduction in medication use as descriptive or exploratory outcomes. The cardiometabolic health improvements observed in our trial are in line with HbA1c, weight loss, and changes in lipids in previous trials led by primary care nurses (Morris et al.) and physicians (Goday et al.). Taken together, our findings suggest similar efficacy for treatment outcomes in low-carbohydrate, energy-restricted nutritional interventions implemented in community pharmacies.

- Lines 138-141, 203-204: During the intervention period pharmacists were in weekly contact with study participants. Is this indeed feasible in clinical practice, within a real-world setting?

Community pharmacists are available and accessible to patients in Canada, making this theoretically possible. If this intervention is efficacious and effective, we feel a compensation/remuneration model could be developed to make this feasible in the real-world and possible in pharmacy practice. We are working on this in follow-up studies. Furthermore, as described above, the lifestyle coaches employed at the pharmacies enhance feasibility.

- Lines 166-167: Shouldn't this approach be combined with lifestyle coaching in order to achieve sustainable changes in diet and physical activity?

We agree that this could be a viable approach to combine a pharmacist-led approach with lifestyle coaching (e.g., by a trained lifestyle coach, diabetes educator, registered dietitian, or kinesiologist). Testing this was beyond the scope of this trial, which was to test the community pharmacist-led approach, focused mainly on medication management.

Reviewer comments, second round –

Reviewer #2 (Remarks to the Author):

The authors have satisfactorily address the reviewer comments which has significantly improved the quality of the paper.

At the editor's discretion this paper would make a useful contribution to the field.

Reviewer #3 (Remarks to the Author):

The authors have addressed all comments in a satisfactory manner. I have no further comments or requested revisions.

Reviewer #4 (Remarks to the Author):

The authors have addressed the points raised previously by reviewer 1. A few minor comments are included below:

- Table 1: Provide units for waist circumference
- Lines 94 and 97: Denote difference as "absolute" difference in the text
- Line 96: Include a sentence in the text on why an HbA1c <6.5% is a clinically meaningful threshold

We have provided a point-by-point response to each reviewer comment in blue type. Relevant excerpts from the revised manuscript are in red font.

REVIEWERS' COMMENTS

Reviewer #2 (Remarks to the Author):

The authors have satisfactorily address the reviewer comments which has significantly improved the quality of the paper.

At the editor's discretion this paper would make a useful contribution to the field.

Reviewer #3 (Remarks to the Author):

The authors have addressed all comments in a satisfactory manner. I have no further comments or requested revisions.

Reviewer #4 (Remarks to the Author):

The authors have addressed the points raised previously by reviewer 1. A few minor comments are included below:

- Table 1: Provide units for waist circumference

Addressed, units for waist circumference have been added to Table 1 and Table 2.

- Lines 94 and 97: Denote difference as "absolute" difference in the text

Addressed, the text in question now reads “absolute difference”.

- Line 96: Include a sentence in the text on why an HbA1c <6.5% is a clinically meaningful threshold

Addressed, the following statement has been added:

(i.e., below the diagnostic threshold for diabetes diagnosis)